# Hygrothermal Characteristics of Cold Roof Cavities in New Zealand

Stephan H. Rupp *[ID], Stephen McNeil, Manfred Plagmann and Greg Overton

Building Research Association of New Zealand, Porirua 5381, New Zealand; steve.mcneil@branz.co.nz (S.M.); manfred.plagmann@branz.co.nz (M.P.); greg.overton@branz.co.nz (G.O.)

\* Correspondence: Stephan.Rupp@branz.co.nz

**Abstract:** The New Zealand Building Code contains minimum durability requirements for components. For roof structures the requirement is 50 years if the component is structural or 15 years if it is not. Metal roof claddings are very common in New Zealand, and roof spaces are typically not deliberately ventilated. Recently, a number of roofs are failing to meet their durability requirement, and the lack of deliberate ventilation is a contributory factor in some cases. In this paper, we consider roof failures and analyse them using the hygrothermal simulation software WUFI® 2D (version 4.1). Using the National Research Council of Canada's *Guideline on Design for Durability of Building Envelopes*, we evaluate to what extent the guideline can be used for such more complex models. Experimental data from a residential dwelling where excessive roof moisture issues were discovered shortly after occupancy are presented. A novel remedial solution using daytime-only ventilation to the roof cavity was trialled, and the data were used to benchmark a two-dimensional numerical simulation of the roof space using WUFI® 2D. A larger hygrothermal data set for 71 dwellings is presented together with relevant climatic conditions. The study works towards evidence-based building code changes for roof ventilation and is an example of using the guideline document for more complicated building envelope assemblies.

**Keywords:** roof space; roof ventilation; WUFI; mechanical ventilation; hygrothermal

## 1. Introduction

The New Zealand Building Code (NZBC) is performance-based. It states how a building must perform in its intended use rather than describing how the building must be designed and constructed. From a durability perspective, compliance requires a practitioner to demonstrate that materials will remain functional for the minimum periods specified (5, 15 or ≥50 years), depending on the criticality and accessibility of the building element. In this paper, we are concerned with roof spaces, and therefore, the materials, components and construction methods must last 50 years if they are structural and 15 years if they are not. To demonstrate that a planned construction will comply with the NZBC, the applicant can use Acceptable Solutions [1], which are specific construction methods that are deemed to comply with the NZBC, or Verification Methods, which are methods of testing or calculation that, if passed, are deemed to comply.

Traditionally, New Zealand roofs have been constructed as so-called cold roofs where the thermal insulation is placed directly above the ceiling lining, leaving a cavity in between the roof cladding and the lining. The volume of this cavity can vary significantly depending on the roof and ceiling design. Conventional truss-based roofs have large roof spaces, whereas skillion-type ceilings leave a very small and often inaccessible roof cavity [2]. Unlike other countries, roof spaces would not be 'deliberately' ventilated in New Zealand—there is no code requirement. Where roof ventilation is specified as part of a building code internationally, it is often done by stating the ratio of free vent area to insulated ceiling area. This would typically be achieved by installing vent openings around the eaves and the ridge of a roof. In North America, ratios between 1:150 and 1:600 are given, with the

value of 1:300 being common [3]. New Zealand has a generally moist, marine climate [4], and whilst roof space failures are not a widespread phenomenon, anecdotal evidence of moisture issues in new dwellings has triggered renewed interest in the role of roof ventilation. In recent years, BRANZ researchers have tried to clarify the role roof space ventilation plays—for example, when is it beneficial, how much is needed or can it cause other problems such as increased corrosion rates of metal fixtures inside the roof space [5]. This work has involved visiting a number of failed roof systems to understand the nature of the problem, and these case studies have also permitted the collection of real-world data and the opportunity to trial solutions.

All cold roof designs, passively ventilated or not, have in common that the roof cavity is hygrothermally coupled to the outdoor climate although by varying degrees. The thermal envelope of a dwelling does not include the roof space. Although the roof cavity temperature will follow the ambient temperature, extreme values above and below the ambient may be observed during daytime heating of the enclosed space and night-time cooling, where energy is radiated into the sky. The degree of coupling between the roof cavity and the outside climate depends on the air exchange rate between these environments. The type of roof cladding, roofing underlay and the detailing around the perimeter are important factors in determining the roof envelope air leakage under natural driving forces. Experimental studies to characterise the air permeability of roof cavities using blower-door or tracer gas methods have been described by Roppel et al. [6] and Rupp et al. [7]. The main drivers for ventilation are wind action around the building and pressure differences due to the stack effect.

While the roof space climate is coupled to the indoor climate of a building, thermal conductive losses across the ceiling during the heating season can impact the temperature in the roof space, the effect is weak due to the ceiling level insulation. If the ceiling level is not an effective air barrier, convective losses are added to the conductive thermal losses. In particular, the movement of inside air into the roof space through any openings at the ceiling level such as old-style downlights, roof access hatches or air-leaky linings may transport significant amounts of water vapour into the roof space [8]. The air permeability of some common ceiling linings and penetrations has been documented by Rupp and Plagmann [9]. The moisture content in the living spaces is typically higher as compared to the outside because human activities release moisture into the building. If this moisture is not removed adequately by ventilation, it raises the risk of issues elsewhere in the structure [4]. This is particularly evident where a high indoor moisture load exists in combination with an easy path for this moisture to be transported into the roof space. This has been regarded as a significant cause of roof cavity moisture issues [10]. Problems can also arise during or shortly after the building process when materials with a high moisture content are closed in too early. Roof moisture issues in New Zealand have been described by Rupp and Plagmann [11,12] and others internationally [6,8,13]. Problems in NZ roofs have typically been ascribed to convection processes. Diffusion-driven moisture transfers are not prevalent and, dedicated vapour control layers are not typically part of a NZ cold roof assembly.

The purpose of roof space ventilation in New Zealand's climatic environment is primarily moisture management, while preventing overheating is less of a concern at present. Moisture entering the roof space, whether from the living spaces below or from the outside, can potentially condense on cold cladding and underlay surfaces, especially during clear nights where the cladding temperature can fall significantly below ambient temperatures due to radiative heat losses into the open sky. Excessive amounts of water may accumulate over time if they are not removed by adequate drying conditions e.g., high enough temperatures to move stored and condensed water into the gas phase and ventilation to subsequently remove it from the roof cavity. However, applying ventilation is not a one-size-fits-all solution. There is potential that the outside air with its natural water content may be a significant moisture source for roof spaces especially in cool marine climates [6]. Too many air exchanges, especially during the night when the cold roof

deck can condense water, may be problematic if enough drying does not occur during the daytime.

In this paper, we present data from a case study where a solar fan was installed as an intervention to help dry out a failing roof space of a new dwelling. These data were then used to benchmark a WUFI® 2D model of the roof space. This model includes ventilation, and captures the absorption and release of moisture from the roofing materials. We also present data from a wider range of dwellings in New Zealand to provide some context on the general potential for deliberate roof ventilation. Particular interest is given to the drying potential of outside air during a daily cycle. The hygrothermal modelling was performed with reference to the National Research Council of Canada's *Guideline on Design for Durability of Building Envelopes* [14]. That document is intended for designers and practitioners (as opposed to researchers) to use when considering a building assembly. The roof spaces modelled in this work represent more complicated models than those covered by the guideline document—two-dimensional versus one-dimensional and including airflows. However, many of the same principles apply, particularly with respect to the selection of boundary conditions.

## 2. Materials and Methods

This section describes the methodologies applied to an individual case study of a newly built residential dwelling with significant roof moisture problems. An intervention to rectify the problems quickly by introducing a daytime only ventilation approach is also described. In addition, a two-dimensional WUFI® 2D simulation of the studied roof space is presented, which aimed to reproduce the observed moisture loads measured in the roof space. Some novel methods are presented, such as incorporating air exchange mechanisms from a mixed source to WUFI® 2D. The final part of this section describes the collection of temperature and humidity data from the roof spaces of 71 households throughout New Zealand over a period of several months showing typical moisture dynamics [15].

### 2.1. Experimental Case Study—New Plymouth, New Zealand

Shortly after occupation of a newly built residential dwelling in New Plymouth, roof moisture issues were discovered with mould growing on the roofing underlay and timber structure in the cold roof space. This led to an experimental case study where hygrothermal data were collected in various parts of the building over several weeks during the southern hemisphere summer of 2019/20. The dwelling, close to sea-level, is a stand-alone single-storey residential building with a 113 m² floor area (143 m² including the garage) on a concrete slab. The walls are mostly constructed using light timber-framing techniques with a brick veneer cladding. The roof is a monopitch design with a low slope of 3°. The high side faces north towards the southern hemisphere sun. A common cold roof construction is adopted with long-run steel cladding of trapezoidal profile over a roofing underlay and support netting. The roof cavity has a volume of approximately 73 m³ and is accessible. Initially, no passive or active ventilation elements were installed to the roof space. The ceiling lining is standard plasterboard and square-stopped, which generally constitutes a good air barrier between the living space and the roof cavity. The ceiling was penetrated by several downlights, but they were of the newer generation, which are tight-fitting, restricting airflow into the roof cavity. The thermal insulation (R3.2) was placed directly above the ceiling, leaving the roof cavity as a space thermally coupled to the outside.

Hygrothermal data were collected for 8 weeks during the summer months of 2019/20. Remote logging sensors recorded temperature and relative humidity in the main part of the living room, bedroom, bathroom and roof cavity (two sensors, which gave identical readings to within their measurement uncertainty). The sensors were SHT15 by Sensirion, with a typical temperature and relative humidity accuracy of ±0.3 °C and ±2.0%, respectively. The sensors were calibrated using a traceable instrument (Vaisala MTI41) just before being installed. Readings were taken approximately every 15 min. The outside climate conditions were obtained from a New Zealand Meteorological Service weather station

10 km from the dwelling (WMO ID: 93309). Meteorological recordings are taken following common WMO practice—i.e., wind speed readings are taken at a reference height of 10 m.

Given the problematic situation in the roof space with its extensive mould growth, a low-cost, effective and nonintrusive method to dry out the roof space was desirable. After the initial monitoring of the as-is situation, a solar-driven fan was installed to provide daytime only extract ventilation to the roof cavity. A SolarStar RM1600 was characterised for free-air and back-pressure performance under a range of irradiances before installation to estimate the required ventilation inlet sizes. At 750 W/m$^2$ solar irradiance, the free airflow is approximately 760 m$^3$/h with a reduction in airflow of approximately 1.1% per Pascal of back pressure. Four 125 mm cowl openings were installed as inlet vents. The fan was installed close to the ridge of the roof while the inlet openings were located around the lower eaves. The actual ventilation rate of the installed fan as a function of time and irradiance could not be measured in this experiment.

### 2.2. Two-Dimensional WUFI® Modelling

We have used WUFI® 2D [16] to create a simplified, two-dimensional numerical model of the roof construction to help understand the nature of the problem and also to ascertain if the guideline document covers all necessary steps to compile such a model. The rectangular geometry of the roof space model has been adapted to represent a cross-section of the average roof space height of the monopitch roof design. The main geometric considerations relevant to a realistic model are the volume and surface area of the installed timber, roofing underlay, insulation and plasterboard ceiling layer (Figure 1). The roof cladding itself is a key factor to correctly capture the significant temperature swings that the roof space experiences, whether this is by solar gain or undercooling.

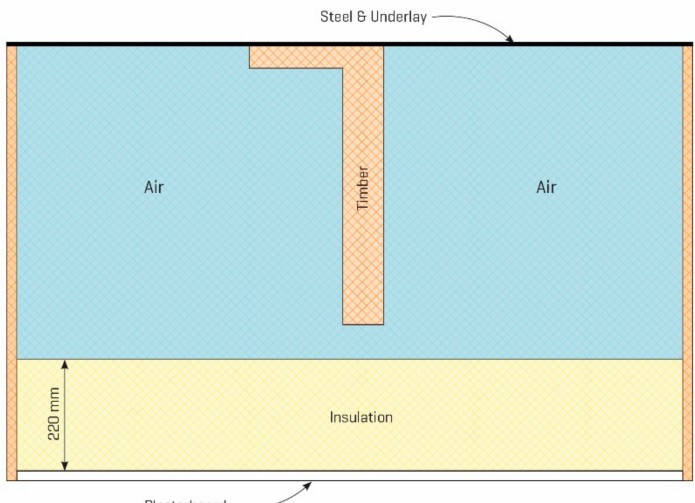

**Figure 1.** Geometry of the two-dimensional WUFI® model of the roof space. A representative timber purlin is in contact with the cladding. The timber rafter/support structure is situated centrally in the air layer. The model is closed by thermal insulation and plasterboard at the bottom and two adiabatic side boundaries.

The hygrothermal properties of the materials in the roof structure are critical to the accuracy of the model. The WUFI® materials database contains generic or commercially available building products. However, the coverage of typical New Zealand manufactured products is limited. Care was taken to choose materials with suitable hygrothermal properties that are representative of those used in the construction. For this study, we have supplemented various properties with materials measured in the BRANZ laboratory. The correct choice of air layers is also critical, and as per usual WUFI® practice, if a layer was ventilated, it was chosen from the database as a layer 'without additional moisture storage'.

Standard air layers exhibit significant inertia, which would obscure the effect we were wanting to simulate.

The boundary conditions are the next important factor to consider. The generic weather data that ships with WUFI do not give satisfactory results as the external conditions are not representative of the local conditions during the measurement campaign. Those used were the external climate conditions measured at a Meteorological Service weather station approximately 10 km from the site and the conditions in the living space of the dwelling. The various surface transfer coefficients were adjusted to reflect the built construction and coatings, and the orientation of the roof was rotated to 3° from horizontal.

The model was then run with various ventilation configurations to test the effect of ventilation on the system. The ventilation sources were from one of three locations—indoors, outdoors and a mix of the two. Details of how the mixed source was implemented are given below. It was deemed impractical to simulate the highly variable (and short timescale) nature of natural ventilation, so we have simply used constant ventilation rates to capture the average effect. In the case where the fan was active, the base ventilation rate was boosted to reflect the fan performance once the global radiation exceeded 50 W/m$^2$. The details of each ventilation source are listed in Table 1.

**Table 1.** Ventilation rates and source of roof cavity air used in the WUFI® 2D modelling.

| Ventilation Rate [ach] | Source of Air |
|---|---|
| 1 | Internal boundary |
| 1 | External boundary |
| 3 | External boundary |
| 3 boosting to 10 | External boundary |
| 1 | 50:50 mix of internal and external air |

The air exchange from a mixed source presented a challenge to apply, and while it can be achieved using the application programming interface (API) available for WUFI® developers, the method described here is accessible to general users. Under normal usage the ventilation air in a WUFI® model will come either from the indoor or outdoor boundary. There is no provision to apply a climate different to these as a ventilation source.

The following describes a method to work around this limitation. The first step needed is to define a climate file, using data measured at location, that describes the actual temperature and humidity of the air that will be used for ventilating the cavity. This can be done in the usual way by using the WAC creation spreadsheet that ships with WUFI®, performing a mass balance for the two air sources and deriving new temperature and relative humidity values. In this case, the climate file was a simple 50:50 mix of the indoor and outdoor air.

To make this new climate file available as a source for ventilation air, it needs to be applied as a boundary condition of the model. Ideally, it should not influence (or perturb) the results of the model apart from its function as a ventilation source. This can be achieved by blocking the flow of heat and moisture through this boundary.

In this case study, the climate file was then applied to a small section of boundary adjacent to the adiabatic guard area. To minimise the impact of this extra boundary condition on the rest of the model, the surface heat transfer coefficient and diffusion thickness were reduced to 0 and increased to 1000 m respectively. This ensured that the flow of heat and moisture was effectively eliminated at this point, but the temperature and relative humidity associated with the boundary were available to ventilate the cavity. Figure 2 shows the location of this extra boundary condition in the model. The blue line is the interior boundary, the black is an adiabatic guard boundary and the red element is the location of the boundary associated with the ventilation source.

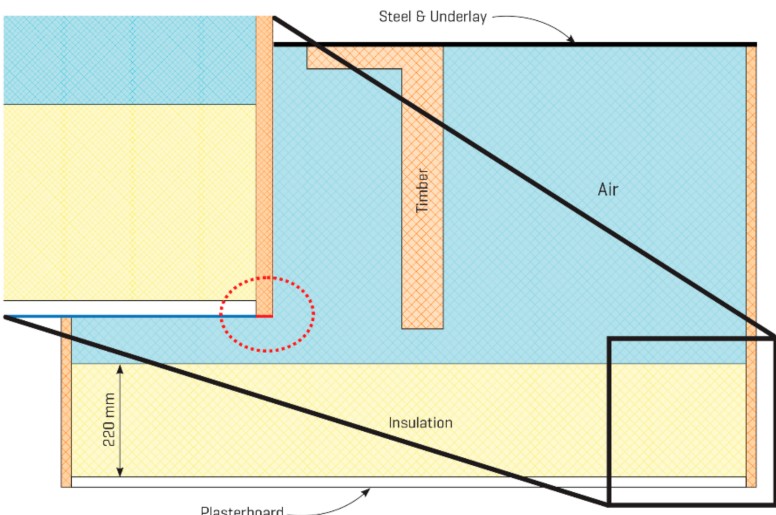

**Figure 2.** Section of the WUFI® 2D geometry. The artificial boundary (red element) is used to provide a distinct external climate for the ventilation air. Heat transfer and diffusion thickness parameters have been set such that there is no flow across this boundary. The 50:50 mix of internal/external air was applied to this surface.

A particularly important physical phenomenon when modelling roofing systems is the undercooling effect. It is a well-known fact that surfaces can fall to temperatures below ambient through radiative energy losses into a clear night sky. Vertical surfaces have less exposure to the sky (in a perfect example, only 50% of the hemisphere), which becomes even less when vegetation, other structures and local terrain features are considered. This means that accounting for the effects of radiation is even more important when considering a roof structure. For the undercooling effect to be incorporated into the simulation, accurate data on atmospheric counter-radiation processes is needed. Energy that is emitted from terrestrial sources, like the roof cladding, is partially reflected from any clouds present in the sky. Even on cloudless nights, the atmospheric air layers will reflect some of this energy back to Earth. WUFI® can account for these processes by specifying the cloud cover in units of octans in the climate files. Unfortunately, cloud cover is not always recorded by meteorological stations.

Early simulations using the cloud cover index only as an input to the model, instead of a comprehensive fully discretised set of radiation data (including the long-wave reflected component), have yielded some poor temperature results for the roof cavity (not shown here). The sensitivity of some models to the radiation balance is discussed in the WUFI® help files in topic 52 Long-Wave Radiation Exchange. To improve the model, we have used Formula (19) in [17] to convert the cloud cover index into counter-radiation data. Different models to determine long-wave counter-radiation from clear sky conditions have also been compared in [18]. Using a discretised set of radiation data has significantly improved the match between the modelled and measured temperature of the roof space for this case study. This is evidence that simulations of roof details require more accurate radiation data than vertical elements such as wall assemblies, reinforcing the messaging in the WUFI® help files. Reference to the increased complexity of roof simulations could be added to future revisions of the guideline [14] document.

With the modelling able to capture the large temperature swings in its diurnal cycle, the effect on the temperature of the underlay and timber support structure in direct contact with the cladding can be simulated. The timber purlins are represented in the model by a single timber structure with a cross-section that gives approximately the correct surface areas in contact with the cladding.

The results of the modelling are detailed in Section 3.2, along with a discussion on limitations of the method and additional opportunities to improve the models.

### 2.3. Housing Survey Data

In 2018/19, BRANZ undertook the Pilot Housing Survey (PHS)—a national housing assessment survey—in partnership with Statistics New Zealand [15]. In addition to the housing assessment survey, a sample of 82 PHS households consented to have temperature and humidity sensors installed in their home for 12 months. This sample of households was recruited at random from households surveyed from November 2018 to May 2019. It was not intended to be representative of the New Zealand housing stock. A subset of 71 dwellings had the roof cavities monitored as well. Table 2 show the sample counts by region.

**Table 2.** Number of dwellings according to the regions across New Zealand used in the PHS study. Not all dwellings had data gathered in the roof cavity.

| Region | Count (Roof Cavities) |
| --- | --- |
| Auckland | 7 (6) |
| Bay of Plenty | 13 (12) |
| Canterbury | 10 (10) |
| Gisborne | 6 (6) |
| Hawke's Bay | 5 (2) |
| Manawatu-Wanganui | 11 (10) |
| Marlborough | 1 (1) |
| Otago | 9 (8) |
| Southland | 5 (5) |
| Taranaki | 3 (2) |
| Tasman | 2 (2) |
| Wellington | 10 (7) |
| TOTAL | 82 (71) |

The sensors used to measure temperature and relative humidity are identical to the ones described in Section 2.1. Up to five sensors were installed inside the dwelling and one on the exterior of the dwelling.

In 71 dwellings, the roof space was accessible via a hatch, and one sensor was placed inside the roof cavity. Potential mould growth was evaluated visually only. The exterior sensor was encased in a perforated plastic container for weather protection and affixed to a wall on the south side of the dwelling, away from direct sunlight and under shelter from the elements where possible (such as under the eaves). In this paper, we consider the measured roof conditions in these 71 dwellings to investigate the potential for extra roof ventilation. No analysis of the link with housing condition is presented.

### 3. Results

Section 3.1 describes the experimental hygrothermal data that was obtained in a newly built residential dwelling in New Plymouth, New Zealand. The impact of the implemented remedial action using a solar-powered ventilator on the hygrothermal condition in the roof space cavity is presented.

In Section 3.2, the obtained experimental data are used to benchmark a two-dimensional numerical model using WUFI® 2D.

Motivated by the findings of the individual case study, a larger data set from the Pilot Housing Study (PHS) is explored. The hygrothermal data obtained in the roof cavities are presented in Section 3.3.

### 3.1. Case Study Results—New Plymouth

The roof space of the case study residential dwelling in New Plymouth is shown in Figure 3. The roofing underlay as well as the timber structure is affected by extensive mould growth.

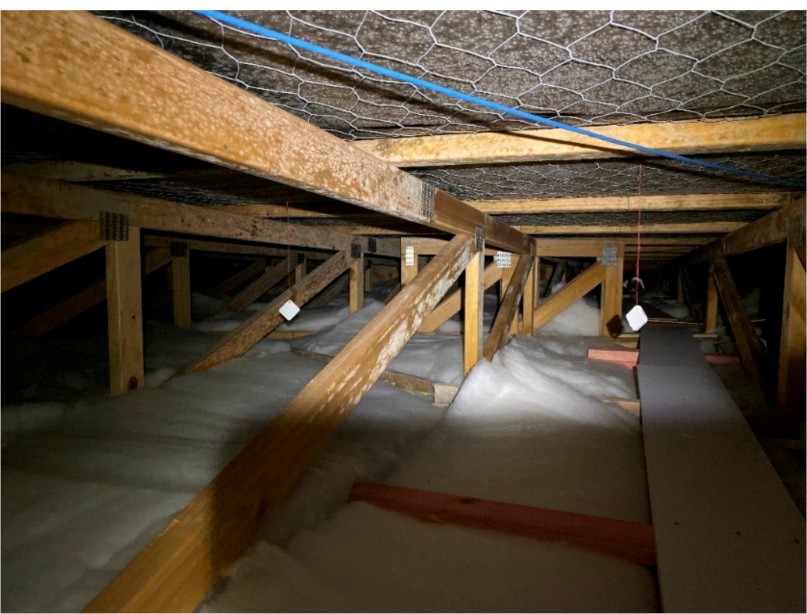

**Figure 3.** Roof space of the case study residential dwelling. Extensive mould growth is visible on the roofing underlay and the timber structure. Two hygrothermal sensors have been mounted in mid-air.

Figure 4 shows the recorded air temperatures in the dwelling during the 2019/20 southern hemisphere summer period. The temperatures in the roof cavity, the living room and the outside are given. The vertical dashed line on 24/01 marks the installation of the solar-powered fan to the roof cavity. As an extract fan, it operates during daytime only with its efficiency coupled to the solar irradiance.

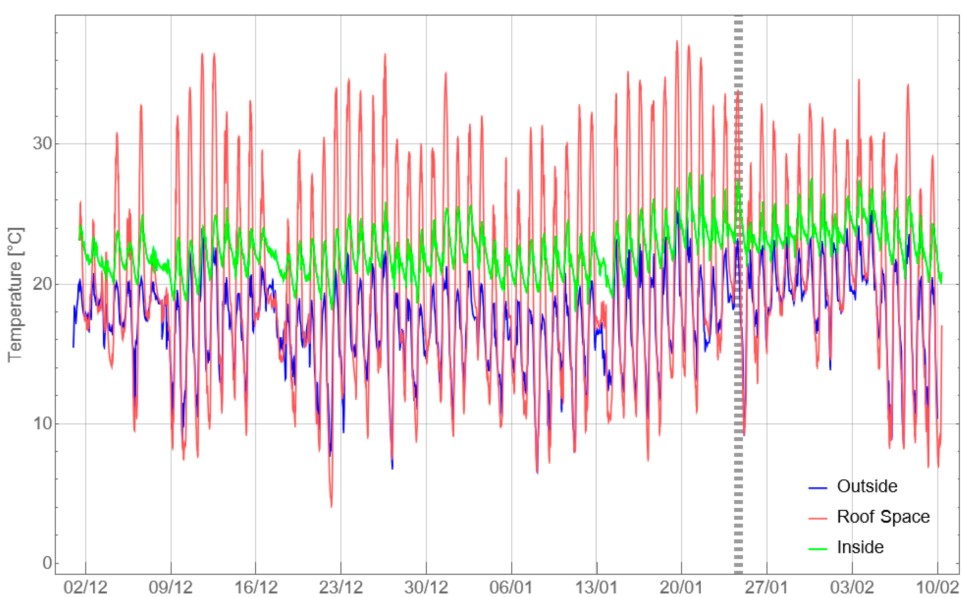

**Figure 4.** Temperature data from the case study dwelling during the southern hemisphere summer. The outside, roof cavity and living room data are given. The vertical dashed line represents the installation of the daytime only roof ventilator.

As expected, the temperatures in the roof space increase significantly above the ambient temperatures during the daytime with solar gains. A number of overcast periods are evident (8/12, 17/12, 13/01) where the roof cavity temperatures are not increasing significantly above the ambient values. Without any air-conditioning, the inside temperature cycles with the outside temperature. During the night-time, the roof space air temperature is largely at ambient. This is true for the periods before and after the fan installation.

Figure 5 shows the water-air mixing ratio expressed as kilograms of water in a kilogram of dry air for the same time period.

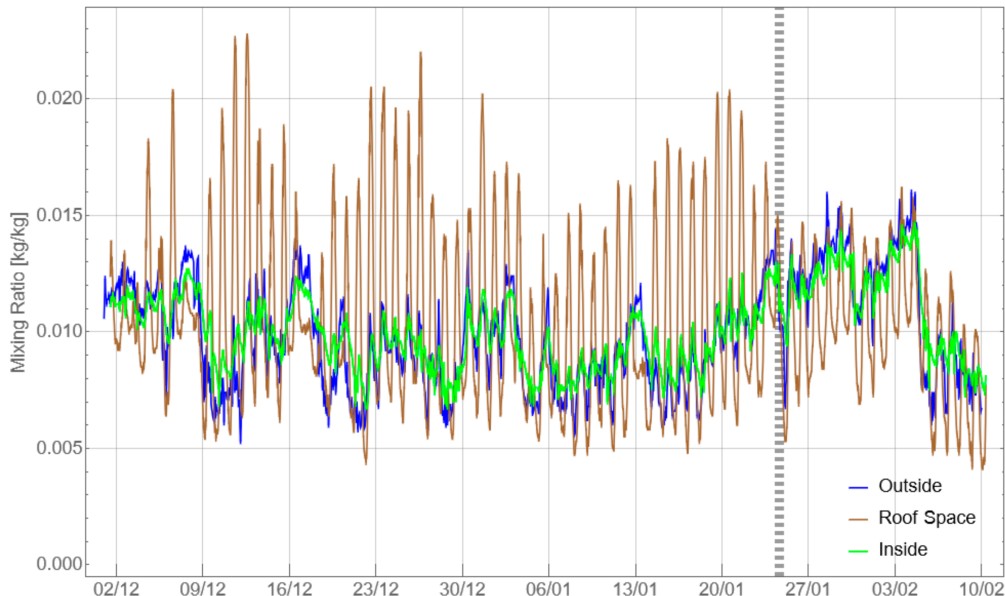

**Figure 5.** Water/air mixing ratio data as in Figure 4. The vertical dashed line represents the installation of the daytime only roof ventilator, showing the removal of the excess moisture from the roof space.

Noticeable are the peaks in mixing ratio in the roof cavity during the daytime, correlating also to the days with elevated temperatures. There are several days where the moisture content in the roof space exceeds 0.020 kg/kg. In contrast, the indoor moisture content is largely between 0.007 and 0.012 kg/kg with a much closer correlation to the outside climate. During the night-time, the moisture content in the roof space air generally drops to a level much closer to ambient. A dramatic effect can be seen after installation of the solar-powered extractor fan. The daytime peaks in the roof space air are significantly reduced, and the climate follows closely the ambient moisture content. During the night-time, the moisture content often falls clearly below the ambient.

Figures 6 and 7 illustrate the impact of the fan in the form of histograms which show daytime and night-time data respectively. The probability of encountering a mixing ratio in steps of 0.001 kg/kg for both the outside and the roof space climates before and after installation of the solar fan is given on the right with the corresponding time series given on the left. Daytime is here defined as the hours between 6:00 and 20:00.

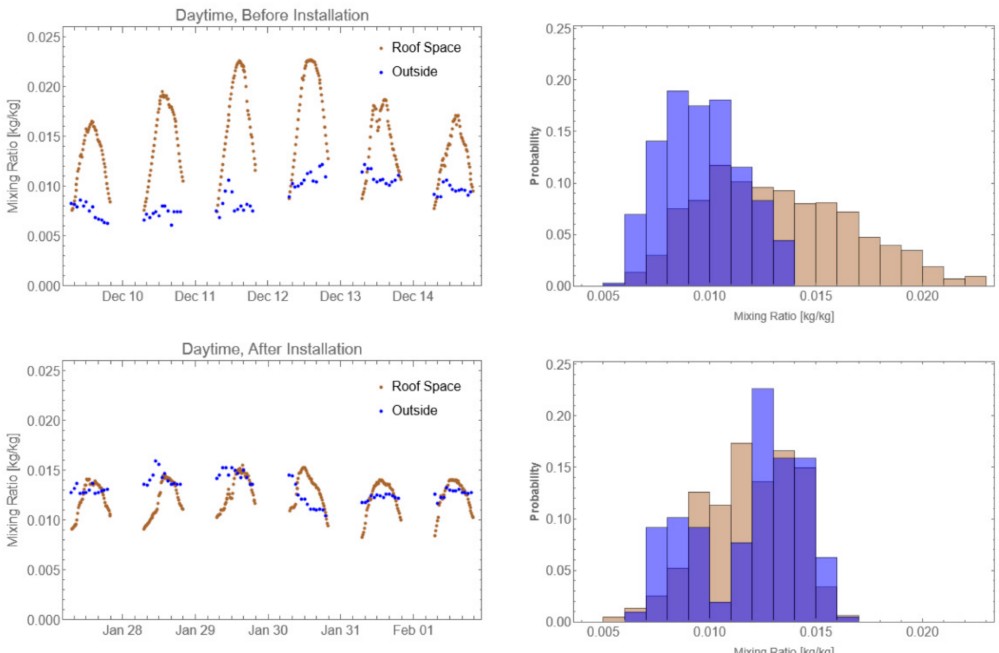

**Figure 6.** Probability histogram (right column) of moisture (mixing ratio [kg/kg]) distribution during daytime hours before (**top**) and after (**bottom**) installation of the solar-driven ventilator. Brown represents the roof cavity data, blue the outside climate. The graphs on the left are the corresponding time series (subset only). After the fan installation, the high moisture end with values above 0.016 kg/kg has been largely removed.

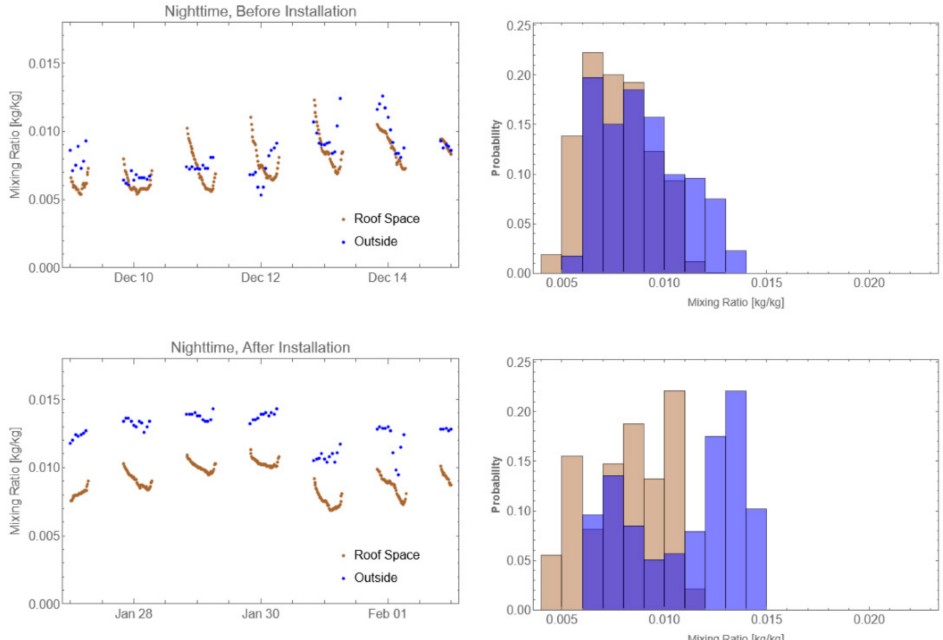

**Figure 7.** Probability histogram (right column) of the moisture distribution during night-time hours before (**top**) and after (**bottom**) installation of the solar-driven ventilator. Brown represents the roof cavity data, blue the outside climate. The graphs on the left are the corresponding time series (subset only).

The daytime probability distribution clearly shows the impact of the solar fan in reducing the absolute moisture content in the roof space cavity. The high moisture load during the day with values greater than 0.016 kg/kg has been removed completely by the active ventilation.

Figure 7 shows the corresponding results for the night-time period, again before and after the fan installation. Since the fan is not operational during the night, any additional air exchanges to the roof space are only due to the passive ventilation properties of the idle fan. For both periods before and after installation of the solar driven fan, the absolute moisture content of the roof space falls below that of the outside during most nights. As expected, there is no significant difference other than the outside climate being more humid during the first 12 days of this measurement period, showing in the probability bar graph. The reason for the drop in water vapour in the roof cavity air might be that the moisture is absorbed by the building components—i.e., the timber or the roofing underlay. The increasing temperature during the day may release this moisture again into the vapour phase. This will be further explored in the WUFI® modelling.

No concluding comments can be made here regarding the source of the excess moisture in the roof space causing the significant mould growth. Given the recent completion of the dwelling, residual construction moisture seems a likely possibility. Occupant behaviour has also been demonstrated to play a role in determining roof moisture levels [10]. Inadequately ventilated living spaces leading to high indoor moisture levels in combination with an air-leaky ceiling can lead to problems. However, Figure 5 shows that, at least during this case study period, the indoor moisture levels are comparable to the outside levels and consequently there are no gains to be had from increasing the air exchanges with the outside.

This case study confirms the effectiveness of daytime-only ventilation in removing moisture from a roof space, in this case, with a solar-driven fan. During the night-time, ventilation should be minimised to avoid an influx from outside moisture into the roof cavity, which might condense on surfaces and get absorbed by the building materials. This is of relevance in particular for roof spaces, where the roof cladding can fall to temperatures below the ambient through radiative energy losses into a clear night sky. The drying potential of ventilating the roof space with outside air during the course of a day is further illustrated by Figure 8. Data are from before the fan installation only.

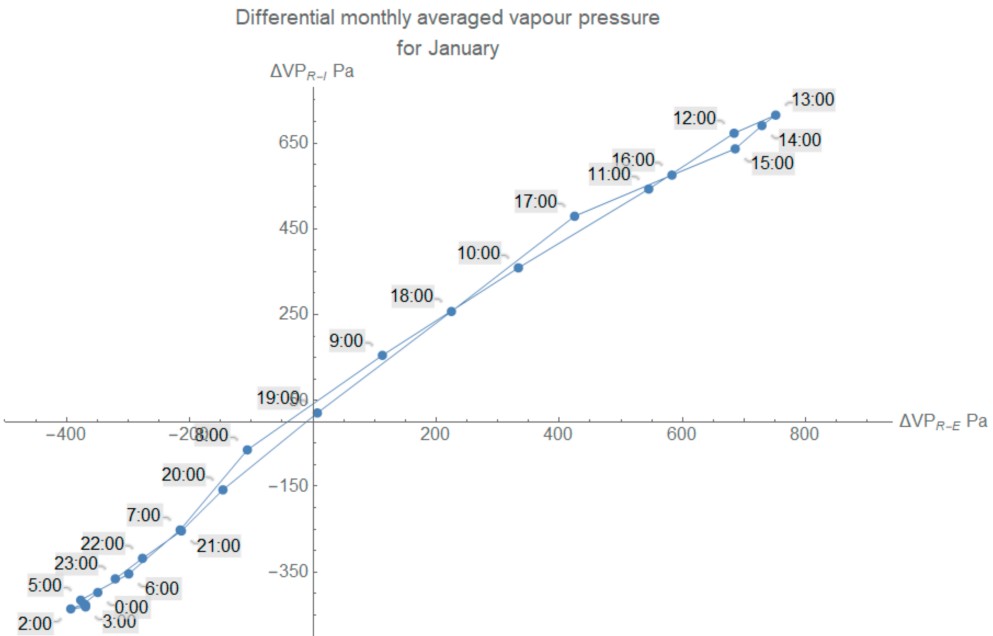

**Figure 8.** Averaged hourly water vapour pressure difference between the roof cavity and the inside ($\Delta VP_{R-I}$) and the external climate ($\Delta VP_{R-E}$) for the New Plymouth case study. Recordings in quadrant I (**top right**) mean that the roof cavity water vapour pressure is higher than the outside and the inside. Quadrant III (**bottom left**) has the opposite situation.

For each daily hour, the two vapour pressure differences are averaged over all days in the monthly period. The data point call-outs denote the hour of the day in local standard time. The *X*-axis is the difference between the roof and the exterior vapour pressure while the *Y*-axis corresponds to the difference between the roof and the interior vapour pressure. Quadrant I describes a climatic condition in which the roof vapour pressure is higher than the interior and the exterior vapour pressure. Air exchange with either environment will therefore reduce the roof vapour pressure. The daytime measurements clearly fall within this quadrant—moisture may be driven out of the materials and into the roof space air by higher temperatures. Conditions in quadrant IV are such that air infiltration from the interior of the building should be avoided as it will increase the vapour pressure in the roof cavity while the exterior air still has drying potential. No data points fall within this quadrant for this case study period, confirming the good indoor conditions—no excess moisture seems to be building up during the daytime, probably a result of good indoor ventilation practice during the summer month. Quadrant III shows data points where the interior and the exterior air has a higher vapour pressure than found in the roof cavity. Measurements during the night-time fall within this segment. A possible explanation is that moisture is condensing on cold surfaces and being absorbed by the building materials, such as the roofing underlay or the timber itself. Moisture can be absorbed into the materials or can form hanging droplets on the surface. A moisture flux into and out of the timber during a daily cycle is confirmed by the WUFI® 2D simulations. Looking at Figure 6 before installation of the ventilator, the mixing ratio increases by approximately 0.010 kg/kg during daytime. By taking the roof cavity volume and the density of air into account, this translates to approximately 1 litre of water that is released from and absorbed again by the materials.

### 3.2. WUFI® Modelling

The plot of the measured and modelled temperatures of the cavity air from the WUFI® 2D simulations are captured in Figure 9 for a period before the solar fan being installed and after.

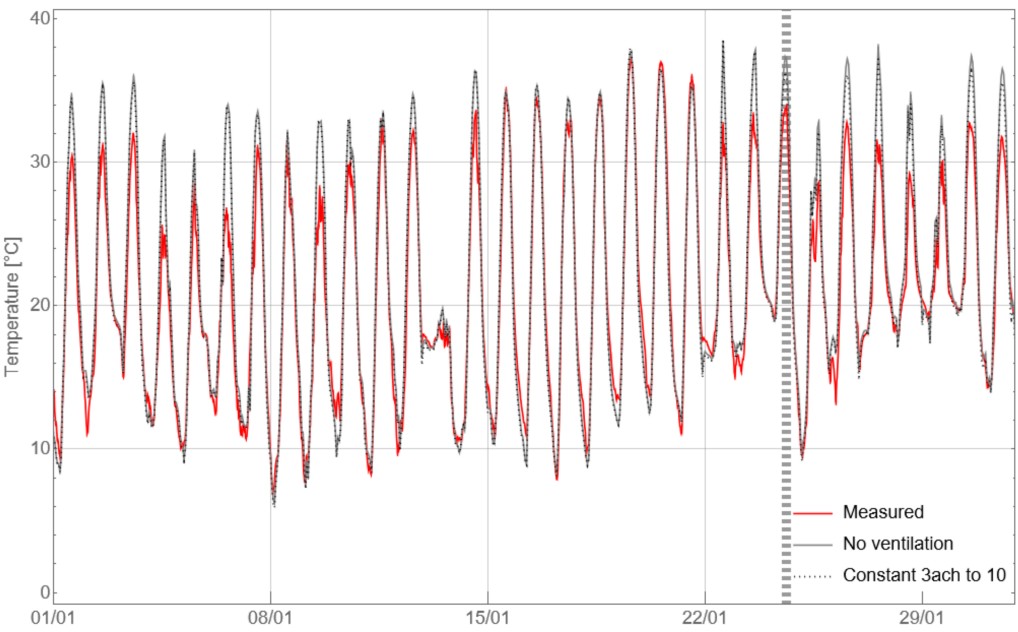

**Figure 9.** Results from the WUFI® 2D simulation—temperature of the roof cavity air simulated for no ventilation and ventilation boosting from 3 ach to 10 ach (daytime, 3 ach otherwise) after the fan is installed.

As described in more detail in the methodology section, two simulation runs are compared to the measured data—no ventilation and a constant background air exchange

of 3 ach, which is boosted to 10 ach by the solar fan when the solar radiation exceeds $50 \, \text{W/m}^2$. The installation is marked by the dashed bars.

For the night-time, we find a very good agreement between the model and the measured data, which gives confidence in the way that the undercooling effect is handled and the assumed background ventilation rate. For most of the daytime periods, we also see a good agreement between the model and the measurements. This is true for some hotter as well as some cooler time periods, especially for the week between 15/01 to 22/01 where the model agrees very well with the measured temperatures. There are several days, however, when the model overestimates the measured temperatures. A possible explanation could be differences in the actual irradiance between the weather station and the dwelling, which are several kilometres apart. It is also a possibility that the effective heat transfer coefficient of the cladding is varying with wind speed. This has not been explored.

Figure 10 shows the relative humidity for the same set of simulations. The plot clearly illustrates the importance of accounting for ventilation process. Without any air exchanges to the roof space, the simulated relative humidity is significantly higher than the measured values. Conversely, the simulation incorporating ventilation has a much better agreement to the measurements.

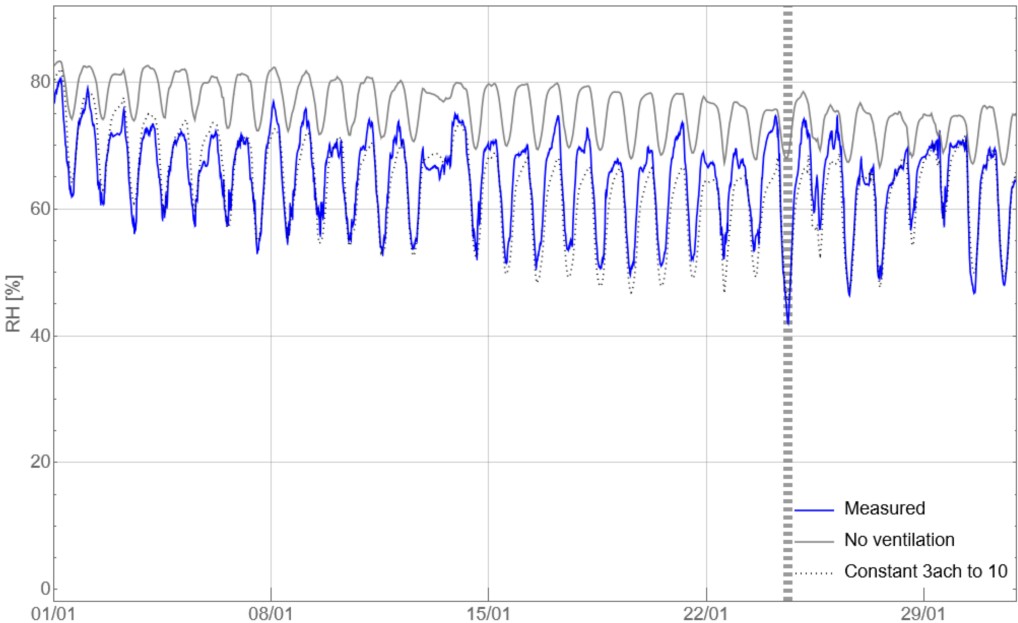

**Figure 10.** Results from the WUFI® 2D simulation—relative humidity of the roof cavity air simulated for no ventilation and ventilation boosting from 3 ach to 10 ach (daytime, 3 ach otherwise) after the fan is installed.

While this may be the case, during the week where we see a very good temperature agreement (15/01–22/01) the relative humidity is slightly underestimated by the model, indicating that more moisture is part of the system—either more water is stored in the materials or it is more readily released. This could come down to several factors. It is thought that this is mainly due to the amount of timber in the system or even the solar absorption/emission coefficients for the roof cladding.

If we change metric and look at the absolute moisture content (mixing ratio) of the cavity air from the WUFI® 2D simulations (Figure 11), the pronounced spikes of moisture during the hotter days are reproduced. By enabling recording of the moisture flux vectors in the results file, the absorption and re-emission of moisture into and out of the wood was clearly visible using the analysis tool 'wufi2dmotion'. Again, without any ventilation processes in the model, these peaks would be significantly overestimated.

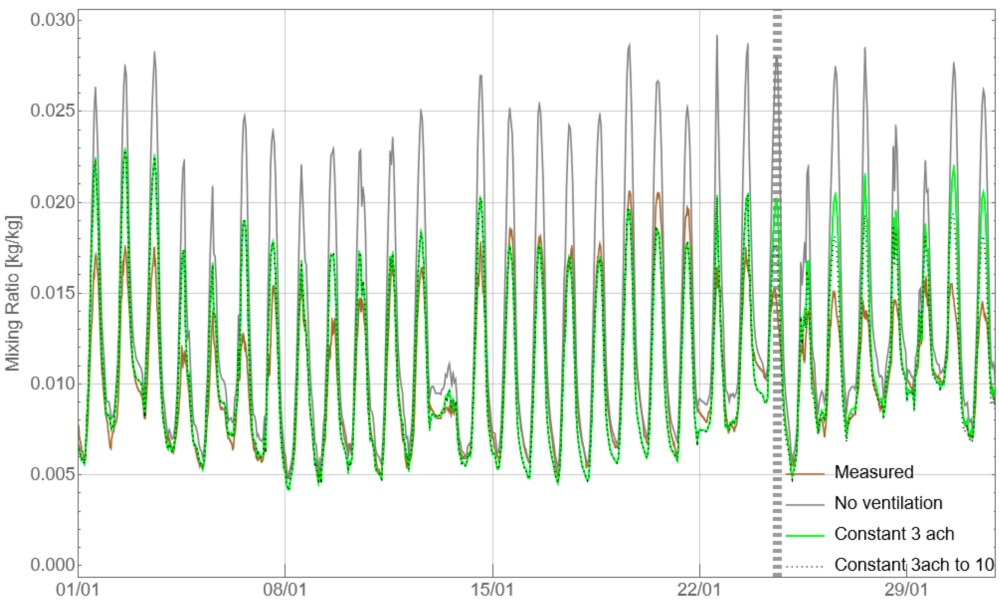

**Figure 11.** Results from the WUFI® 2D simulation—mixing ratio of the roof cavity air simulated for no ventilation and ventilation boosting from 3 ach to 10 ach (daytime, 3 ach otherwise) after the fan is installed. The graph also shows the constant ventilation rate of 3 ach only, illustrating the difference to the fan.

The results demonstrate that the model can capture the effect as observed in the measurements. However, there are limitations to the models, which are discussed below. Introducing ventilation processes to the model reduces the relative humidity and the mixing ratio values as observed in reality. The additional boosting of the ventilation rate from a background value of 3 ach to 10 ach has resulted in a further decrease in the absolute moisture values, although not as low as was observed in reality. This is likely tied to the amount of storage capacity in the model. However, subtle but significant effects such as the influence of the additional ventilation on the boundary layer air in contact with the roof cladding and underlay are not possible to incorporate. These effects are likely to have an influence on the amount of evaporation of condensed moisture but are likely difficult to measure in a field situation, requiring tightly controlled laboratory measurements to observe. Even with these limitations, the modelling confirms that there is buffering of the moisture in the air into the surrounding materials as the structure cools and releases it again when the temperature rises.

There is, however, still room to improve the models without major modification, which is the focus of the remainder of the section. To improve the quality of the model fit, there are several approaches that could be taken as both the quality of the boundary condition data and some fundamental limitations with the WUFI® kernel are likely having an impact. As stated earlier, a constant background ventilation rate was assumed to apply in this instance, as it gave a reasonable agreement. In reality, there is going to be considerable variation due to wind and stack effects, so the question is whether this variation is significant enough to warrant a more complex ventilation source. From an understanding of the physical processes perspective, this is likely the case. However, from a practitioner's perspective trying to assess the durability of a construction methodology, the assumed constant rate is likely to be sufficient, provided it captures the majority of the real-world effect.

The roof surface temperature is strongly dependent on both the incident and reflected radiation. Moving away from cloud cover only to a full explicit radiation and counter-radiation balance as an input to the model has had a positive impact on the simulated results. An issue present is that the dwelling is inland from the weather station so it is likely experiencing greater cloud cover and less wind than the well-exposed site of the weather station on the coast. To improve the irradiation data, which is especially critical to

roof models, the data would need to be captured on site. However, the equipment needed to gather this data is expensive and not suitable for field deployment. It is possible this will be covered in future work using the data acquisition available at BRANZ, which will also enable the use shorter time steps of the order of 10 min (or even less), which will increase the numerical stability of the model.

Given the current 1-h resolution data, it may be possible to apply a convolution similar to that in [19] on the results to smooth response to peaks. The model appears to react very quickly to changing boundary conditions and given the very small amount of moisture in the air relative to the storage in the materials comprising the structure, this is understandable. The technique taken Section 4.5 in [19] has proved successful in understanding the limitations in modelling air infiltration in relatively airtight dwellings. In tandem with the response, there is also a fundamental limitation in how moisture in air is represented in WUFI®. Due to numerical limitations, the moisture-carrying capacity of air adheres to a fixed moisture storage function. Typically, this is that same as that for air at 20 °C. In the context of situations where the temperature of the ventilated space does not have large swings over the course of a day, this is an acceptable approximation. In the context of a cold roof, the diurnal swings in temperature are such that the ability of air to hold moisture can change by an order of magnitude in a short space of time. To address this, an approach like that taken by McNeil [20] with WUFI® 1D could be applied to the 2D model. In this paper, two independent WUFI® 1D models were coupled to each side of a third independent mathematical model of the air. The independent model allowed for improvements to the moisture storage function, and the radiative heat transfer across the cavity. There is a considerable amount of work needed to perform this style of modification, particularly with the 2D version of WUFI®, so it was not done for this work. There is, however, an API for WUFI® 2D that would enable this to be developed, so it remains a possibility for the future. As this work was focused on demonstrating a technique available to any WUFI® user, it will remain a task for future work to try and capture these effects more accurately.

Overall, the WUFI® 2D analysis suggests reasonable agreement with the hypothesis that the daytime peaks in the measured air moisture content arise due to the release of moisture from the building materials. However, the complexity of these types of building elements, where the conditions are dependent on rarely available data sets, suggests that they remain outside of the scope of the guideline document.

### 3.3. Housing Survey Data

In order to understand the dynamics of the ventilation potential (using passive vents as opposed to a solar fan) for a larger data set, we look again at the differences of roof vapour pressure with interior and exterior vapour pressures using the Pilot Housing Study (PHS) as described above. For these houses, the current level of roof ventilation is unknown, so we are assessing the scope for additional ventilation beyond what is occurring already. The graph in Figure 12 shows the differences for each hour averaged over a month for an individual dwelling of the PHS without an obvious moisture problem in the roof space. The results for January are similar to the New Plymouth case study with smaller vapour pressure differences in the first quadrant, however. During the daytime, when the roof cavity temperature is high, the vapour pressure differences are within quadrants I and IV and hence ventilation during this time will reduce the vapour pressure in the roof. During the night, roof infiltration/ventilation should be minimised. Figure 12 also shows the situation for the southern hemisphere winter month of July. The indoor climate has a higher average moisture content than the roof space at all times, and air movement into the roof cavity should be avoided. Ventilation with the outside is still effective during daytime although the drying potential is much reduced in comparison to the summer period.

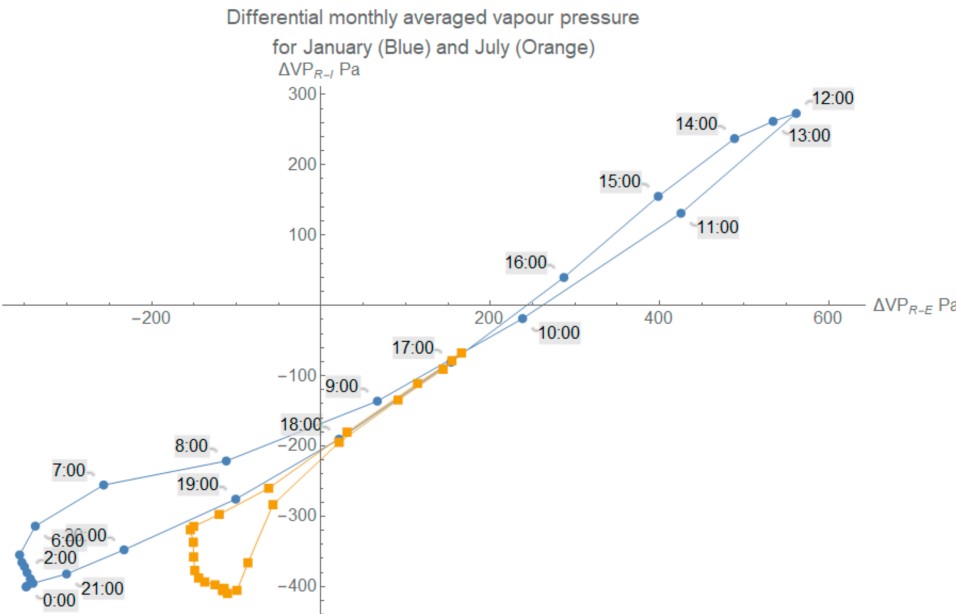

**Figure 12.** Water vapour pressure difference between the roof cavity and the inside ($\Delta VP_{R-I}$) and the external climate ($\Delta VP_{R-E}$) for one of the PHS dwellings and for a summer (blue) and winter (orange) month. Recordings in quadrant I (**top right**) mean that the roof cavity water vapour pressure is higher than the outside and the inside. Quadrant III (**bottom left**) has the opposite situation.

Most roof climates in the study are predominantly operating in quadrants I and IV during the day and III during the night (Figure 13). Only three homes had roof climates that operated for long periods in quadrant II. The data given in Figure 13 are for a subset from the PHS where roof cavity measurements are available. Not all dwellings have had recordings for all time periods. 71% of these dwellings had long-run metal claddings, 12% metal roofing tiles and 17% concrete tiles. Daily readings for all days of the months of January, March, June and November are averaged over the respective month to yield the ellipse representing the daily cycle as in Figure 12. Times have been omitted for clarity. Each colour represents one identical dwelling across all graphs.

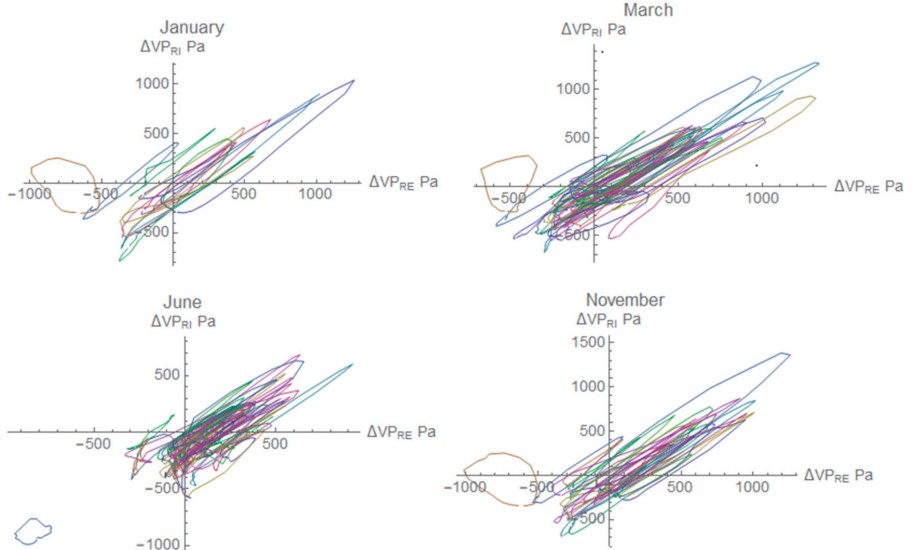

**Figure 13.** Water vapour pressure difference as in Figure 12 for a greater number of dwellings from the Pilot Housing Survey for the months of January, March, June and November.

The seasonal dynamics of the drying potential are illustrated. Some roofs do not deviate much from the average vapour pressure dynamics throughout the year while others see a contraction of dynamics not reaching quadrant I as shown in Figure 12. As a detailed analysis of occupant behaviour was not part of this study, no conclusions can be drawn on the individual differences. However, it is noteworthy that only one of the dwellings in Figure 13 had mould growth problems in the roof space recorded in the survey. The vapour pressure difference curve for this dwelling is well within the other houses and not one of the buildings with larger vapour pressure differences. Also, the vapour pressure differences of the New Plymouth case study house are not on the extreme end of the ones in Figure 13. While this is relevant for removing moisture from storage sources and drying out the roof space, the vapour pressure differences cannot be used to make comments on the occurrence of mould growth itself. The relevant parameter for the latter is relative humidity, which was not part of this study.

While this data set is not a valid statistical representation either across New Zealand or across different roof designs and materials, it nevertheless represents a trend with similar patterns. It illustrates our findings that the drying potential of a roof space is highly dependent on the individual circumstances of the dwelling and the interior and exterior climate. In general, however, infiltration/ventilation of the roof cavity will yield an improvement of the water vapour pressure given the prolonged periods of time in quadrant I. The fact that most houses in New Zealand with a cold roof design do not suffer from moisture-related problems further suggests that the line between having problems and having no moisture-related problems is difficult to establish in a general sense.

The case study data highlighted the importance of daytime ventilation, and this is reinforced by the PHS data. However, passive ventilation openings allow air to enter the roof cavity throughout the entire day. It is therefore important to understand wind speed distribution over the course of a day. Figure 14 shows the wind speed distribution for New Zealand cities Christchurch and Wellington for the winter months of July and August and the spring month of September. Lighter colours represent a higher count in the respective time/wind speed window.

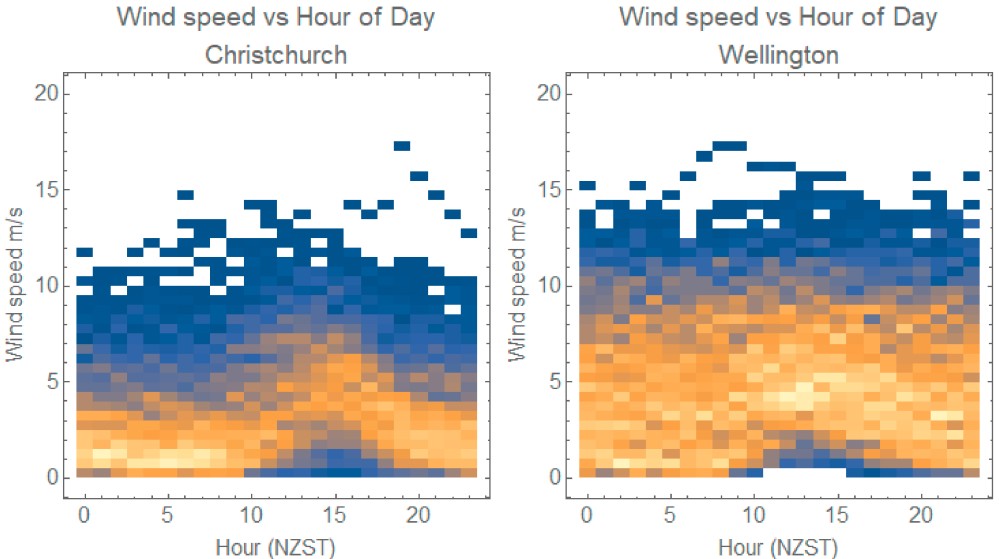

**Figure 14.** Hourly wind speed in metres per second distribution over the course of a day for Christchurch and Wellington averaged for July to September. Lighter colours represent a higher count in the respective time/wind speed window.

For Christchurch, (Figure 14) the distribution shows that during the day the wind speed is often higher than during the night. Ventilation over the course of a day will likely result in a net water vapour pressure reduction in the roof cavity. The results for Wellington

show a more uniform wind speed distribution over the course of the whole day. It is therefore more likely that the drying potential of ventilation is reduced in such a situation.

As the wind speed distribution is different between day and night and the daytime ventilation is the preferred ventilation, it is important to understand what the likelihood is that daytime ventilation exceeds night-time ventilation. This is shown in Figure 15, where the daytime/night-time differential probability is shown for Christchurch and Wellington. The Christchurch data show a clear demarcation between daytime and night-time probability so that the daytime ventilation and drying potential clearly exceeds that of the night-time. The Wellington data show the night-time drying potential probability is somewhat interspersed with the daytime ventilation but the daytime drying potential probability still dominates. This means that there is no increased risk of condensation due to the additional ventilation. Any condensation should not be able to accumulate over long periods of time.

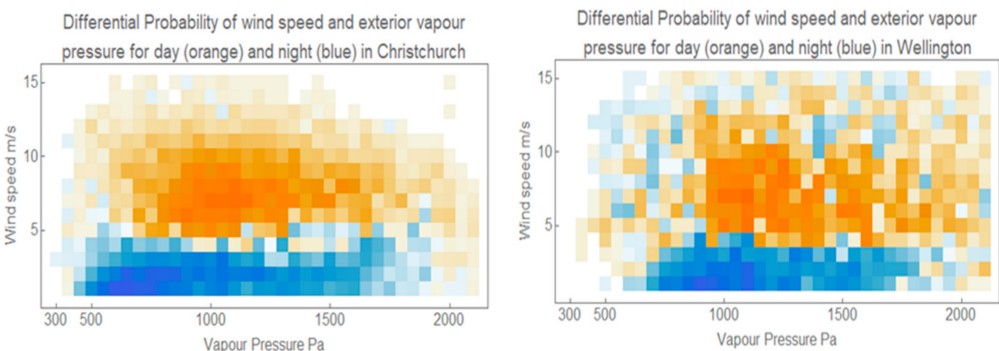

**Figure 15.** Hourly wind speed in metres per second distribution over the course of a day, averaged for July to September. Darker colours represent a higher count in the respective time/wind speed window.

## 4. Discussion

The two-dimensional WUFI® modelling of a roof space undertaken in this work lies outside the scope of the guideline document. This is particularly true given the specialist nature of the work needed to obtain the boundary conditions. It is a good example of the effort needed to simulate a system on the margins of long-term durability, where the 'devil is in the detail', and shows how remedial measures can be arrived at by considering all the available evidence.

To obtain the agreement in the WUFI® model, measured temperature and humidity in the building and local meteorological data were required, both of which are likely to be out of reach of the general practitioner, and not practically available at the design stage. It also required substantial work with respect to radiative effects and absorption/emission coefficients to obtain a good agreement. What this work gives, however, is a baseline case sensitivity analysis for cold roof design. Given this model, a set of methods could be developed to aid in design decisions.

The impacts of climate change were explored using NZ climate forecast models. However, as there is not a cloud cover component to these files at present, it limits the application in this case. For future work they would be a reasonable basis for testing resilience of walls to climate change effects as walls are less sensitive to radiative cooling effects.

A missing aspect that presents a challenge in applying the guideline, is connecting the risk of mould growth to the durability of specific materials in the assembly. In a case like this it could be argued that the mould growth is sufficient to trigger a failure in respect of the 15-year durability clause for the NZBC for the roof cladding and underlay. What is a more pressing concern however is what do roof space conditions like these mean in terms of 50-year durability of the structure itself? Current work at BRANZ is monitoring the corrosion rates of a number of types on fasteners in a variety of timber species/treatment

options. These are fully exposed to the elements, and all relevant parameters are being recorded (weather variables, as well as timber moisture contents, temperatures). One of the longer term aims of this work is to contribute to a corrosion postprocessor for WUFI® that will enable the likelihood of fasteners achieving a service life of greater than 50 years to be assessed.

## 5. Conclusions

In this paper, we have demonstrated how a daytime-only ventilation strategy using a solar-powered fan to the roof cavity has quickly and effectively improved the moisture levels in the cavity air. The original moisture source that has led to the substantial mould growth in the roof space of the case study could not definitively be determined. However, given that there is no evidence of excessive indoor moisture levels moving across the ceiling into the roof space, initial high construction moisture levels of this new dwelling seem to be the likely cause.

The experimental data from the case study, before and after the fan intervention, have been simulated by a two-dimensional WUFI® model yielding reasonably good results. However, this required some important factors to be represented in the boundary conditions and may be beyond the skillset of a general WUFI® practitioner as outlined in the guideline document. The observed absolute moisture peaks in the cavity during the daytime at elevated temperatures are consistent with a source/sink mechanism where moisture is released during the day and absorbed again during the night.

The acquired climate data of the case study roof space have been analysed to yield a novel way of illustrating the highly time-dependent drying potential of roof space air. Our findings illustrate that the drying potential of a roof space is dependent on the individual circumstances of the dwelling and the interior and exterior climate. Using results from a wider set of houses, this potential for daytime ventilation has been demonstrated. Infiltration or designed ventilation of the roof cavity will generally yield an improvement of the water vapour pressure in the roof space given the prolonged periods of time in quadrant I, where the vapour pressure is higher in the roof space than the outside. When using passive vents, night-time ventilation will also occur, which is generally less desirable. However, we have demonstrated a way of assessing the likelihood of beneficial outweighing detrimental aspects. The fact that most roof spaces of houses in New Zealand do not suffer from moisture-related problems suggests that the line between having problems and having no problems is a fine one.

Design guidelines for durability may be more useful in underpinning generic solutions such as warm roofs, rather than relying on them to perfectly predict the behaviour of individual roofs such as those described in this paper. This application of a guideline would facilitate a new range of solutions for practitioners to use that do not require individual analysis. It is likely that the New Zealand Building Code will require some step changes to meet climate change obligations and those step changes potentially permit a move away from traditional construction practices to new systems which have inherently larger margins of safety. Design guidelines would provide a mechanism for verifying that these solutions still meet the durability requirements of the New Zealand Building Code.

**Author Contributions:** Conceptualisation, methodology and data curation, M.P., S.H.R., S.M. and G.O.; experimental investigation, S.H.R.; formal analysis, M.P., S.M. and S.H.R.; writing—original draft preparation, S.H.R. and G.O.; funding acquisition, S.H.R. and M.P. All authors have read and agreed to the published version of the manuscript.

**Funding:** This research was funded by the New Zealand Building Research Levy, grant number MR0021—Roof Ventilation Risk Analysis.

**Data Availability Statement:** Data are stored on local databases and servers at BRANZ Ltd., Porirua, New Zealand.

**Acknowledgments:** Thanks to Vicki White for making the Pilot Housing Study data available.

**Conflicts of Interest:** The authors declare no conflict of interest. The funders had no role in the design of the study, the collection, analysis or interpretation of data, the writing of the manuscript or the decision to publish the results.

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
