# Peer review of "Hygrothermal Characteristics of Cold Roof Cavities in New Zealand"

_buildings, doi:10.3390/buildings11080334_

Round 1

Reviewer 1 Report

Interesting investigation and comparison between modelling and experimental data.

Author Response

Thank you for your work and your positive conclusion. Much appreciated

Reviewer 2 Report

Dear author,

my comments are in a pdf file.

Best regards

Author Response

Thank you for your feedback - much appreciated. The comments and requests are certainly valid and have improved the paper. We have addressed each comment in the pdf document.

Reviewer 3 Report

Authors presented results from the case study with a solar fan to dry a failing roof space of a building. The drying potential of ambient air during the day is analysed. The paper has a traditional structure, including problem description in the Introduction. Some sections should be revised. Some comment to improve the manuscript are given below:

The abstract is too long. The abstract should be a total of about 200 words maximum.

Add some quantitative results into Abstract in order to have a better view of your paper.

Only three keywords are used as the minimum. Probably you could revise and add some more keywords.

Figure’s numbering in the text should be consistent (line 135).

Accuracy should be provided for the measurement equipment.

Figure 1: some dimensions could be recommended.

The title of Figure 1 could be shortened.

Figure 2 is not clear. It should be revised, add some legend.

Figure 3 presents not results; it is more a part of the methodology (line 299).

Revise title of Figure 5. Also, revise the title of y-axis.

What is Roof space 2? (fig. 4., fig. 5).

The quality of Fig. 6 is poor. The same comment for Fig. 7.

Figure 6 should be discussed in deeper. Results must be discussed.

What is standard error or other value comparing modelling and measurement results?

Figure 13: it is difficult to understand it. Recommendation is to select other visualisation type.

Add some quantitative results into Conclusions.

Line 148-149: wind speed was taken at height of 10 m. What was the actual height for the case study?

Author Response

(The authors gave the same response as above.)
